# Interligand Charge-Transfer Processes in Zinc Complexes

**Carlo Ciarrocchi** [1,†] , **Guido Colucci** [1,†], **Massimo Boiocchi** [2] , **Donatella Sacchi** [1], **Maduka L. Weththimuni** [1] , **Alessio Orbelli Biroli** [1,*] and **Maurizio Licchelli** [1,*]

1   Dipartimento di Chimica, Università di Pavia, Via Taramelli 12, 27100 Pavia, Italy;
    carlo.ciarrocchi@unipv.it (C.C.); guido.colucci@unipv.it (G.C.); donatella.sacchi@unipv.it (D.S.);
    madukalankani.weththimuni@unipv.it (M.L.W.)
2   Centro Grandi Strumenti, Università di Pavia, Via Bassi 21, 27100 Pavia, Italy; massimo.boiocchi@unipv.it
*   Correspondence: alessio.orbellibiroli@unipv.it (A.O.B.); maurizio.licchelli@unipv.it (M.L.);
    Tel.: +39-0382-987-936 (M.L.)
†   These authors contributed equally to the work.

**Abstract:** Electron donor–acceptor (EDA) complexes are characterized by charge-transfer (CT) processes between electron-rich and electron-poor counterparts, typically resulting in a new absorption band at a higher wavelength. In this paper, we report a series of novel 2,6-di(imino)pyridine ligands with different electron-rich aromatic substituents and their 1:2 (metal/ligand) complexes with zinc(II) in which the formation of a CT species is promoted by the metal ion coordination. The absorption properties of these complexes were studied, showing the presence of a CT absorption band only in the case of aromatic substituents with donor groups. The nature of EDA interaction was confirmed by crystallographic studies, which disclose the electron-poor and electron-rich moieties involved in the CT process. These moieties mutually belong to both the ligands and are forced into a favorable spatial arrangement by the coordinative preferences of the metal ion.

**Keywords:** coordination chemistry; zinc complex; 2,6-di(imino)pyridine ligand; electron donor–acceptor complex; charge transfer process; UV-visible absorption spectroscopy; X-ray structure





## 1. Introduction

The interaction between electron donor (D) and electron acceptor (A) species, providing well-defined adducts generally indicated as electron donor–acceptor (EDA) complexes, [1] has been known for a long time [2–6]. EDA complexes are also commonly reported as charge-transfer (CT) complexes, as the interaction between the electron-rich and the electron-poor counterparts involves a charge-transfer process. The electronic spectra of EDA (or CT) complexes typically show the appearance of a new absorption band at lower energies, which is characteristic of neither D nor A but is due to a CT transition [7,8]. Although the characteristics of CT processes have been extensively investigated along the course of the last few decades [5,8–11], a renewed interest has been more recently addressed to CT compounds due to their application in different research areas, [12,13] particularly in the field of organic functional materials (e.g., superconductors, [14] semiconductors [15,16], and ferroelectrics [17]).

An important role of CT processes in biological systems has been also described so far [18] and more recently, some CT adducts have been investigated for their antimicrobial activity [19–22] and for possible application in drug science [23–27].

CT processes have been also exploited to develop several examples of colorimetric and/or fluorescent sensors and probes for different substrates, including compounds displaying biological and pharmacological relevance [28–30] and chemosensors [31,32].

Several fascinating supramolecular assemblies and devices (for instance catenanes, rotaxanes and host–guest systems) have been prepared by exploiting the interactions between donor and acceptor components [33–41].

It has also been reported that CT complexes may be stabilized also by the contribution of additional interactions, such as, for example, hydrogen–bond [42–44]. However, only a limited number of examples have been reported where the formation of EDA adducts is fostered by metal–ligand interactions [45–50]. This effect exerted by the coordination of a metal ion can be explained mainly on the basis of steric or electronic effects. In fact, the metal ion can properly steer the interaction of molecular fragments, for instance by inducing a suitable folding of a coordinating subunit where the D and A components are bound, and at the same time allowing them to approach each other [45,46]. Moreover, the complexation of metal ions can strongly affect the electron density of the molecular components (e.g., aromatic moieties) involved in the formation of the EDA adduct. For instance, ligand subunits such as pyridine [47] or terpyridine [48] undergo a decrease in their electron density upon coordination; thus, they can act as acceptor subunits and properly interact with the corresponding donor counterparts.

Previously [51], we reported the formation of intracomplex CT species following the coordination of ligands based on the 2,6-di(imino)pyridine subunit (see Figure 1, compounds **1a** and **1b**) to metal ions such as Zn$^{II}$ and Cd$^{II}$ according to a 1:2 (metal/ligand) stoichiometry. The EDA interactions involving naphthalene subunits belonging to one ligand molecule and the complexed 2,6-di(imino)pyridine fragment of the other one are favored both for electronic and steric reasons: (i) the metal ion coordination lowers the electron density of the di(imino)pyridine moieties which become electron-deficient and interact with the electron-rich naphthalene fragments; (ii) the complementary aromatic subunits on each ligand molecule are forced by the metal ion coordination geometry (octahedral) into a spatial arrangement extremely favorable to donor–acceptor interactions [51].

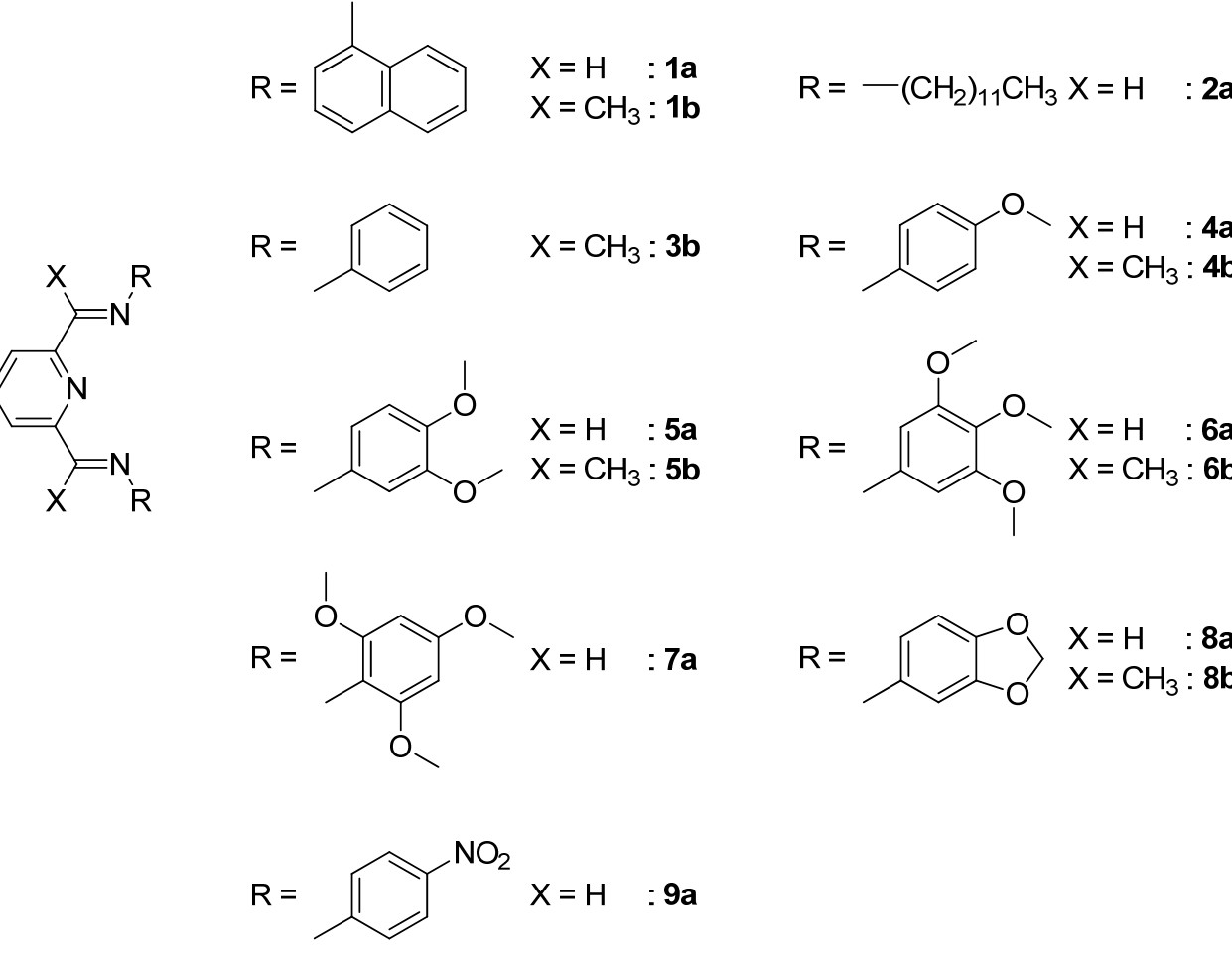

**Figure 1.** Structures of the di(imino)pyridine derivatives investigated.

In this paper, we report the results of a more systematic study performed on a family of compounds (**3**–**8**) in which the 2,6-di(imino)pyridine-chelating unit is conjugated to different electron-rich aromatic substituents. In particular, the absorption properties of zinc(II) complexes formed by **3**–**8** are examined in order to confirm the formation a CT species promoted by the metal ion coordination. The behavior of these compounds is also compared to that of ligands **2** and **9** which do not have the right structural features to afford intramolecular CT species.

## 2. Materials and Methods

### 2.1. General Remarks

Unless otherwise stated, commercially available reagent-grade chemicals were used as received; solvents were used without purification.

Spectrophotometric grade solvents were used for UV-vis measurements.

UV-vis spectra were recorded on a Hewlett-Packard 8453 diode array or on a Varian Cary 100 spectrophotometer.

NMR spectra were recorded on a Bruker AMX400 spectrometer.

Spectrophotometric titrations were performed on 10 mL samples of solutions of the ligand ($10^{-5}$ M) in $CH_3CN$, by microadditions of $CH_3CN$ stock solutions of zinc(II) triflate. In each experiment, the overall addition was limited to about 400 µL, so that volume increment variation was not significant.

Titration data were processed with HypSpec software (Hyperquad suite) to determine the equilibrium constants [52,53].

### 2.2. Synthesis of Ligand

General Procedure

The synthesis of **1a** and **1b** has been already reported [51]. All di(imino)pyridine derivatives (except **7a** and **9a**) were prepared according to the following general procedure: solid 2,6-diacetylpyridine or 2,6-diformylpyridine was mixed with the envisaged neat amine under magnetic stirring at room temperature. In a few minutes, the mixture gave a clear solution and then a white slurry. A small amount of methanol (2–3 mL) was added and the stirring continued for 12 h. The resulting suspension was filtered under vacuum and the solid product washed with methanol. No further purification was necessary for the di(imino)pyridine ligands obtained by this procedure. Experimental details for each preparation are reported below.

**2a**: 1-dodecylamine (0.275 g, 1.48 mmol) and 2,6-pyridinecarboxaldehyde (0.1 g, 0.74 mmol) reacted according to the general procedure. A slight heating was required after reagent mixing, in order to completely melt the amine (MP = 27–29 °C).

Yield: 260 mg (0.55 mmol, 74%). ESI-MS: $m/z$(%) = 470 (100) [M + H]$^+$. $^1$H-NMR (CDCl$_3$): δ = 8.45 (s, 2 H, N=CH), 8.05 (d, *J* = 7.6 Hz, 2H, PyH), 7.82 (t, *J* = 7.6 Hz, 1 H, PyH), 3.70 (t, *J* = 7.0 Hz, 4 H, C=NCH$_2$), 1.72 (m, *J* = 7.0 Hz, 4 H, C=NCCH$_2$), 1.2–1.4 (br m, 36 H, NC$_2$(CH$_2$)$_9$C), and 0.89 (t, *J* = 7.0 Hz, 6 H, CH$_3$). Elemental analysis calculated (%) for C$_{31}$H$_{55}$N$_3$: N 8.94, C 79.26, H 11.80; found N 8.90, C 78.97, H 11.78. Absorption (MeCN) UV-vis: λ$_{max}$ = 286 nm (ε = 6020 M$^{-1}$cm$^{-1}$).

**3b**: benzylamine (1 mL, 9.1 mmol) and 2,6-diacetylpyridine (0.70 g, 4.3 mmol) were reacted according to the general procedure. Yield: 1.12 g (3.3 mmol, 76%); yellow solid. ESI-MS: $m/z$ = 342 (100%) [M + H]$^+$; 364 (20) [M + Na]$^+$. $^1$H-NMR (CDCl$_3$): δ = 8.25 (d, *J* = 8.0 Hz, 2H, PyH), 7.76 (t, *J* = 8.0 Hz, 1H, PyH), 7.49 (d, *J* = 8.0 Hz, 4H, BenzH), 7.39 (t, *J* = 8.0 Hz, 4H, BenzH), 7.30 (d, *J* = 8.0 Hz, 2H, BenzH), 4.82 (s, 4H, CH$_2$), and 2.55 (s, 6H, CH$_3$). Elemental analysis calculated (%) for C$_{23}$H$_{23}$N$_3$: N 12.31, C 80.90, H 6.79; found: N 12.28, C 81.20, H 6.77. Absorption UV-vis (MeCN): λ$_{max}$ = 282 nm (ε = 8000 M$^{-1}$cm$^{-1}$).

**4a**: 4-methoxybenzylamine (0.23 mL, 1.8 mmol) and 2,6-pyridinedicarboxaldehyde (0.1 g, 0.74 mmol) reacted according to the general procedure. Yield: 0.18 g (0.48 mmol, 65%), white solid. ESI-MS: $m/z$ (%)= 374 (100) [M + H]$^+$. $^1$H-NMR (CDCl$_3$): δ = 8.51 (s, 2H, N=CH), 8.09 (d, *J* = 7.8 Hz, 2H, PyH), 7.80 (t, *J* = 7.8 Hz, 1H, PyH), 7.29 (d, *J* = 8.7 Hz,

4H, BenzH), 6.92 (d, *J* = 8.7 Hz, 4H, BenzH), 4.85 (s, 4H, CH$_2$), and 3.83 (s, 6H, OCH$_3$). Elemental analysis calculated (%) for C$_{23}$H$_{23}$N$_3$O$_2$: N 11.25, C 73.97, H 6.21; found N 11.21, C 73.68, H 6.23. Absorption UV-vis (MeCN): $\lambda_{max}$ = 284 nm ($\varepsilon$ = 11960 M$^{-1}$cm$^{-1}$).

**4b**: 4-methoxybenzylamine (1 mL, 7.6 mmol) and 2,6-diacetylpyridine (0.60 g, 3.7 mmol) reacted according to the general procedure. Yield: 1.25 g (3.1 mmol, 84%), pale yellow solid. ESI-MS: *m/z* (%) = 402 (100) [M + H]$^+$. $^1$H-NMR (CDCl$_3$): δ = 8.22 (d, *J* = 8.1 Hz, 2H, PyH), 7.74 (t, *J* = 8.1 Hz, 1H, PyH), 7.39 (d, *J* = 8.4 Hz, 4H, BenzH), 6.93 (d, *J* = 8.4 Hz, 4H, BenzH), 4.75 (s, 4H, CH$_2$), 3.84 (s, 6H, OCH$_3$), and 3.55 (s, 6H, CH$_3$). Elemental analysis calculated (%) for C$_{25}$H$_{27}$N$_3$O$_2$: N 10.47, C 74.79, H 6.78; found: N 10.50, C 74.55, H 6.76. Absorption UV-vis (MeCN): $\lambda_{max}$ = 278 nm ($\varepsilon$ = 9200 M$^{-1}$cm$^{-1}$).

**5a:** 3,4-Dimethoxybenzylamine (0.12 mL, 0.8 mmol) and 2,6-pyridinedicarboxaldehyde (0.05 g, 0.37 mmol) reacted according to the general procedure. Yield: 0.075 g (0.17 mmol, 47%), white solid. ESI-MS: *m/z* (%) = 434 (100) [M + H]$^+$. $^1$H-NMR (CDCl$_3$): δ = 8.52 (s, 2H, N=CH), 8.08 (d, *J* = 8.2 Hz, 2H, PyH), 7.79 (t, *J* = 8.2 Hz, 1H, PyH), 7.15 (s, 2H, BenzH); 6.98 (d, *J* = 8.3 Hz, 2H, BenzH); 6.88 (d, *J* = 8.3 Hz, 2H, BenzH), 4.84 (s, 4H, CH$_2$), 3.95 (s, 6H, OCH$_3$), and 3.90 (s; 6H; OCH$_3$). Elemental analysis calculated (%) for C$_{25}$H$_{27}$N$_3$O$_4$: N 9.69, C 69.27, H 6.28; found: N 9.72, C 69.03, H 6.31. Absorption UV-vis (MeCN): $\lambda_{max}$ = 287 nm ($\varepsilon$ = 14500 M$^{-1}$cm$^{-1}$).

**5b**: 3,4-Dimethoxybenzylamine (0.5 mL, 3.3 mmol) and 2,6-diacetylpyridine: (0.27 g, 1.65 mmol) reacted according to the general procedure. Yield: 0.440 g (0.95 mmol 58%), white solid. ESI-MS: *m/z* (%) = 462 (100) [M + H]$^+$. $^1$H-NMR (CDCl$_3$): δ = 8.22 (d, *J* = 8.2 Hz, 2H, PyH), 7.75 (t, *J* = 8.2 Hz, 1H, PyH), 7.05 (s, 2H, BenzH), 7.00 (d, *J* = 8.3 Hz, 2H, BenzH), 6.88 (d, *J* = 8.3 Hz, 2H, BenzH), 4.74 (s, 4H, CH$_2$), 3.95 (s, 6H, OCH$_3$), 3.90 (s, 6H, OCH$_3$), and 2.56 (s, 6H, CH$_3$). Elemental analysis calculated (%) for C$_{27}$H$_{31}$N$_3$O$_4$: N 9.10, C 70.26, H 6.77; found N 9.12, C 70.15, H 6.74. Absorption UV-vis (MeCN): $\lambda_{max}$ = 282 nm ($\varepsilon$ = 14800 M$^{-1}$cm$^{-1}$).

**2,4,6-trimethoxybenzylamine**: 2,4,6-trimethoxybenzonitrile (0.2 g, 1.03 mmol) was dissolved in THF (50 mL). LiAlH$_4$ (0.3 g) was slowly added and the resulting suspension magnetically stirred for 24 h at room temperature, under nitrogen atmosphere. Water (10 mL) was then slowly dropped and the resulting mixture filtered. The liquid phase was extracted with CH$_2$Cl$_2$ (3 × 30 mL) and the organic phase desiccated over anhydrous Na$_2$SO$_4$. The removal of solvents afforded a yellow oil. Yield: 140 mg (0.71 mmol, 69%). ESI-MS: *m/z* (%)= 181.1 (35) [M − NH$_3$]$^+$, 197.9 (20) [M + H]$^+$, and 394.8 (100) [2M + H]$^+$. No further purification was necessary and the product was directly used in the synthesis of compound **6a**.

**6a**: 3,4,5-trimethoxybenzylamine (0.26 mL, 1.52 mmol); 2,6-pyridinecarboxaldehyde (0.1 g, 0.74 mmol) reacted according to the general procedure. Yield: 204 mg (0.41 mmol, 56%). ESI-MS: *m/z* (%) = 494 (20) [M + H]$^+$, 516 (100) [M + Na]$^+$. $^1$H-NMR (CD$_3$CN): δ = 8.52 (s, 2 H, N=CH), 8.10 (d, *J* = 7.7 Hz, 2 H, PyH), 7.90 (t, *J* = 7.7 Hz, 1 H, PyH), 6.71 (s, 4 H, BenzH), 4.80 (s, 4 H, CH$_2$), 3.83 (s, 12 H, OCH$_3$), and 3.73 (s, 6 H, OCH$_3$). Elemental analysis calculated (%) for C$_{27}$H$_{31}$N$_3$O$_6$: N 8.51, C 67.51, H 6.33; found N 8.49, C 67.25, H 6.34. Absorption UV-vis (MeCN): $\lambda_{max}$ = 280 nm ($\varepsilon$ = 12860 M$^{-1}$cm$^{-1}$).

**6b**: 3,4,5-trimethoxybenzylamine (0.26 mL, 1.52 mmol) and 2,6-diacetylpyridine (0.12 g, 0.74 mmol) reacted according to the general procedure. Yield: 224 mg (0.43 mmol, 58%), white solid. ESI-MS: *m/z* (%) = 522 (100) [M + H]$^+$. $^1$H-NMR (CDCl$_3$): δ = 8.21 (d, *J* = 8.1 Hz, 2 H, PyH), 7.89 (t, *J* = 8.1 Hz, 1 H, PyH), 6.70 (s, 4 H, BenzH), 4.72 (s, 4 H, CH$_2$), 3.83 (s, 12 H, OCH$_3$), 3.73 (s, 6 H, OCH$_3$), and 2.53 (s, 6H, CH$_3$). Elemental analysis calculated (%) for C$_{29}$H$_{35}$N$_3$O$_6$: N 8.06, C 66.78, H 6.76; found: N 8.04, C 66.59, H 6.80. Absorption UV-vis (MeCN): $\lambda_{max}$ = 280 nm ($\varepsilon$ = 10300 M$^{-1}$cm$^{-1}$).

**7a**: 2,4,6-trimethoxybenzylamine (140 mg, 0.71 mmol) was dissolved in 10 mL of methanol. The solution was magnetically stirred and 40 mg of solid 2,6-pyrdinecarboxaldehyde (0.30 mmol) was added. A white solid formed over a period of about 2 h; stirring was continued for 24 h and then the white product was collected by filtration.

Yield: 0.103 g (0.21 mmol, 70%), white solid. ESI-MS $m/z$ (%): 494 (100) [M + H]$^+$; 987 (15) [2M + H]$^+$. $^1$H-NMR (CDCl$_3$): δ = 8.30 (s, 2 H, N=CH); 8.05 (d, $J$ = 7.6 Hz, 2 H, PyH); 7.74 (t, $J$ = 7.6 Hz, 1 H, PyH); 6.18 (s, 4 H, BenzH); 4.90 (s, 4 H, CH$_2$); and 3.83 (m, 18 H, OCH$_3$). Elemental analysis: calculated (%) for C$_{27}$H$_{31}$N$_3$O$_6$: N 8.51, C 67.51, H 6.33; found N 8.54, C 67.32, H 6.36. Absorption UV-vis (MeCN): λ$_{max}$ = 287 nm (ε = 10400 M$^{-1}$cm$^{-1}$).

**8a**: piperonylamine (0.10 mL, 0.8 mmol) and 2,6-pyridinedicarboxaldehyde (0.050 g, 0.37 mmol) reacted according to the general procedure. Yield: 0.070 g (0.17 mmol, 47%); white solid. ESI-MS: $m/z$ (%) = 402 (100) [M + H]$^+$. $^1$H-NMR (CDCl$_3$): δ = 8.50 (s, 2H, N=CH), 8.05 (d, $J$ = 7.8 Hz, 2H, PyH), 7.89 (t, $J$ = 7.9 Hz, 1H, PyH), 6.91 (s, 2H, BenzH), 6.85 (br m, 4H, BenzH), 5.97 (s, 4H, O-CH$_2$-O), and 4.78 (s, 4H, C=NCH$_2$). Elemental analysis calculated (%) for C$_{23}$H$_{19}$N$_3$O$_4$: N 10.47, C 68.82, H 4.77; found N 10.45, C 69.12, H 4.75. Absorption UV-vis (MeCN): λ$_{max}$ = 288 nm (ε = 17100 M$^{-1}$cm$^{-1}$)

**8b**: piperonylamine (0.10 mL, 0.8 mmol) and 2,6-diacetylpyridine (0.06 g, 0.37 mmol) reacted according to the general procedure. Yield: 70 mg (0.17 mmol, 46%); white solid. ESI-MS: $m/z$ (%) = 430 (100) [M + H]$^+$. $^1$H-NMR (CDCl$_3$): δ = 8.18 (d, $J$ = 7.8 Hz, 2H, PyH), 7.82 (t, $J$ = 7.8 Hz, 1H, PyH), 7.02 (s, 2H, BenzH), 6.94 (d, $J$ = 7.9 Hz, 2H, BenzH), 6.85 (d, $J$ = 7.9 Hz, 2H, BenzH), 5.97 (s, 4H, O-CH$_2$-O), 4.68 (s, 4H, C=NCH$_2$), and 2.52 (s, 6H, CH$_3$). Elemental analysis calculated (%) for C$_{25}$H$_{23}$N$_3$O$_4$: N 9.78, C 69.92, H 5.40; found N 9.80, C 69.72, H 5.42. Absorption UV-vis (MeCN): λ$_{max}$ = 287 nm (ε = 16100 M$^{-1}$cm$^{-1}$).

**9a**: pyridinedicarboxaldehyde (0.06 g, 0.44 mmol) was added to a methanol solution (10 mL) containing 3-nitrobenzylamine hydrochloride (0.19 g, 1.02 mmol) and of triethylamine (0.15 mL, 1.10 mmol) under magnetic stirring. A white solid formed in one hour; stirring was maintained for 12 h. The solid product was recovered by suction filtration and washed with a few mL of cold methanol. Yield: 100 mg (0.25 mmol, 57%). ESI-MS: $m/z$ (%) = 404 (100) [M + H]$^+$, 426 (25) [M + Na]$^+$. $^1$H-NMR (CDCl$_3$): δ = 8.60 (s, 2H, N=CH), 8.25 (d, $J$ = 8.7 Hz, 4H, BenzH), 8.17 (d, $J$ = 7.8 Hz, 2H, PyH), 7.90 (t, $J$ = 7.8 Hz, 1H, PyH), 7.58 (d, $J$ = 8.8 Hz, 4H, BenzH), and 5.01 (s, 4H, C=NCH$_2$). Elemental analysis: calculated (%) for C$_{21}$H$_{17}$N$_5$O$_4$: N 17.36, C 62.53, H 4.25; found N 17.31, C 62.70, H 4.21. Absorption UV-vis (MeCN): λ$_{max}$ = 279 nm (ε = 28570 M$^{-1}$cm$^{-1}$).

## 2.3. X-ray Crystallographic Studies

Diffraction data were collected at room temperature by using an Enraf–Nonius CAD4 four-circle diffractometer (Enraf-Nonius, Delft, The Netherlands) for [Zn$^{II}$(**5b**)$_2$](CF$_3$SO$_3$)$_2$·MeOH and [Zn$^{II}$(**7a**)$_2$](CF$_3$SO$_3$)$_2$ crystals, whereas a Bruker-Axs Smart-Apex CCD-based diffractometer (Bruker AXS Inc., Madison, WI, USA) was used for the [Zn$^{II}$(**8a**)$_2$] (CF$_3$SO$_3$)$_2$ crystal. Both instruments were equipped with a graphite-monochromatized Mo Kα X-radiation (λ = 0.7107Å). Single crystals of the [Zn$^{II}$(**5b**)$_2$](CF$_3$SO$_3$)$_2$·MeOH complex were unstable outside the solution in which they formed and, to prevent decay, diffraction data were collected with the crystal placed in a closed glass capillary containing a small amount of mother liquor. Crystal data for the studied complexes are reported in Table 1.

Data reductions (including intensity integration, background, Lorentz, and polarization corrections) for reflections collected with the conventional diffractometer were performed with the WinGX software package; [54] absorption effects were evaluated with the psi-scan method [55] and absorption correction was applied to the data. Omega rotation frames (scan width of 0.3°, scan time of 40 sec., and sample-to-detector distance of 6 cm) collected by means of the CCD-based diffractometer were processed with the SAINT software [56] and intensities were corrected for Lorentz and polarization effects. Absorption effects were analytically evaluated by the SADABS software [57] and correction was applied to the data.

Crystal structures were solved by direct methods (SIR 97) [58] and refined by full-matrix least-square procedures on $F^2$ using all reflections (SHELXL-2018) [59]. Anisotropic displacement parameters were refined for all nonhydrogen atoms; hydrogens were placed at calculated positions with the appropriate AFIX instructions and refined by using a riding model. CCDC-2182821, CCDC-2182822, and CCDC-2182823 contain the supplementary

crystallographic data for this paper. These data can be obtained free of charge from the Cambridge Crystallographic Data Centre.

**Table 1.** Crystal data for investigated crystals.

| | [Zn(5b)$_2$](CF$_3$SO$_3$)$_2$·MeOH | [Zn(7a)$_2$](CF$_3$SO$_3$)$_2$ | [Zn(8a)$_2$](CF$_3$SO$_3$)$_2$ |
|---|---|---|---|
| Formula | C$_{57}$H$_{66}$F$_6$N$_6$O$_{15}$S$_2$Zn | C$_{56}$H$_{62}$F$_6$N$_6$O$_{18}$S$_2$Zn | C$_{48}$H$_{38}$F$_6$N$_6$O$_{14}$S$_2$Zn |
| $M$ | 1318.67 | 1350.63 | 1166.35 |
| Color | yellow | pale green | yellow-green |
| Dimension (mm) | 0.72 × 0.58 × 0.43 | 0.50 × 0.40 × 0.05 | 0.70 × 0.50 × 0.20 |
| Crystal system | monoclinic | monoclinic | triclinic |
| Space group | $P2_1/c$ (no. 14) | $P2_1/c$ (no. 14) | $P$-1 (no. 2) |
| $a$ [Å] | 18.737(7) | 10.825 (1) | 10.759(4) |
| $b$ [Å] | 14.179(2) | 26.380(2) | 11.214(3) |
| $c$ [Å] | 24.740(6) | 21.484(2) | 23.274(2) |
| $\alpha$ [°] | 90 | 90 | 87.79(1) |
| $\beta$ [°] | 110.02(2) | 93.79(1) | 87.89(1) |
| $\gamma$ [°] | 90 | 90 | 61.65(2) |
| $V$ [Å$^3$] | 6175(3) | 6121.5(9) | 2468.9(12) |
| $Z$ | 4 | 4 | 2 |
| $\rho_{calcd}$ [g cm$^{-3}$] | 1.418 | 1.465 | 1.569 |
| $\mu$ Mo$_{K\alpha}$ [mm$^{-1}$] | 0.553 | 0.563 | 0.679 |
| Absorption corr. type | psi-scan | multi-scan | psi-scan |
| Min/max transmission | 0.748/0.788 | 0.781/0.972 | 0.712/0.875 |
| Scan type | $\omega$ scans | $\omega$ scans | $\omega$ scans |
| θ range [°] | 1.2–26.0 | 1.2–25.0 | 1.7–30.0 |
| Measured reflections | 12849 | 43639 | 15199 |
| Unique reflections | 12139 | 10721 | 14402 |
| $R_{int}$ | 0.0733 | 0.0270 | 0.0279 |
| Strong data [I$_O$>2σ(I$_O$)] | 3905 | 7834 | 6800 |
| $R1$, $wR2$ (strong data) | 0.1071, 0.2493 | 0.1083, 0.3402 | 0.0719, 0.1656 |
| $R1$, $wR2$ (all data) | 0.2813, 0.3472 | 0.1305, 0.3659 | 0.1657, 0.2105 |
| GOF | 1.021 | 1.606 | 1.014 |
| Refined parameters | 790 | 802 | 694 |
| Max/min residuals [eÅ$^{-3}$] | 0.84/−0.48 | 2.02/−1.03 | 0.71/−0.51 |

## 3. Results and Discussion

### 3.1. Synthesis and Absorption Properties of 2,6-di(imino)pyridine Ligands

2,6-di(imino)pyridine derivatives **2**–**9**, whose structure is depicted in Figure 1, were prepared by the direct reaction of 2,6-diacetylpyridine or 2,6-diformylpyridine with primary amines bearing the envisaged aliphatic/aromatic groups.

The synthesis of all these compounds could be simply carried out in MeOH at room temperature as previously reported for ligands **1a** and **1b** [51], although in some cases, the yields are not satisfactory. Moreover, this experimental procedure often affords not pure products. Di(imino)pyridine derivatives can be alternatively prepared by mixing solid diacetyl- or diformyl-pyridine with the envisaged liquid amine, without using any solvent. This simple "heterogeneous" procedure, which is quite simple and generally provides enough pure products, was adopted to prepare most investigated compounds, that were obtained satisfactorily pure with yields ranging between 47 and 84%. Compounds **7a** and **9a** were prepared by reacting the corresponding amines and pyridine derivatives in methanol solution. The reactions of 2,6-diacetylpiryidine with dodecylamine and 4-nitrobenzylamine (hypothetical products **2b** and **9b**, respectively) and of 2,6-diformylpiryidine with benzylamine (hypothetical product **3a**) were repeatedly carried out by both experimental procedures, always yielding complex mixtures. Any attempts to isolate **2b**, **3a**, and **9b** as pure compounds or to purify the reaction mixtures were unsuccessful.

All the ligands reported in Figure 1 were obtained as white or pale-yellow solids which did not require any special purification prior to further investigations.

The UV-visible absorption behavior of all the examined compounds was investigated in acetonitrile solutions ($10^{-5}$ M concentration). The absorption data are summarized in Table 2.

**Table 2.** Absorption data for compounds **2–9** obtained in MeCN ($10^{-5}$ M).

| Compound | $\lambda_{max}$(nm) | $\varepsilon$(M$^{-1}$cm$^{-1}$) |
|:---:|:---:|:---:|
| **2a** | 286 | 6020 |
| **3b** | 282 | 8000 |
| **4a** | 284 | 11,960 |
| **4b** | 278 | 9200 |
| **5a** | 287 | 14,500 |
| **5b** | 282 | 14,800 |
| **6a** | 280 | 12,860 |
| **6b** | 280 | 10,300 |
| **7a** | 287 | 10,400 |
| **8a** | 288 | 17,100 |
| **8b** | 287 | 16,100 |
| **9a** | 279 | 28,570 |

All spectra show unstructured or poorly structured bands in the 250–320 nm range. For compound **2a,** which contains only aliphatic substituents, the band centered at 286 nm can be ascribed to the di(imino)pyridine π-system. In compounds **3–9,** these bands are similarly centered at about 280 nm but are more intense since they result from the overlapping of the absorptions due to the aromatic chromophores and the di(imino)pyridine π-system (see Table 2). The UV-vis spectra do not change by increasing the concentration of the derivatives; in particular, no absorption bands are observed at a lower energy even at relatively high concentration values (up to $10^{-3}$ M), indicating that compounds **3–9** do not interact intra- or intermolecularly to generate adducts with CT character.

### 3.2. Interaction with Zn$^{2+}$: Spectrophotometric and ESI-MS Investigations

The behavior of systems **2–9** in the presence of metal ions was investigated at first by spectrophotometric titration experiments: a solution of the envisaged ligand in MeCN ($10^{-5}$ M) was titrated with zinc(II) trifluoromethanesulfonate and an absorption spectrum was acquired after each addition.

Absorption spectra registered during the titration of **2a** with zinc(II) trifluoromethanesulfonate are reported in Figure 2.

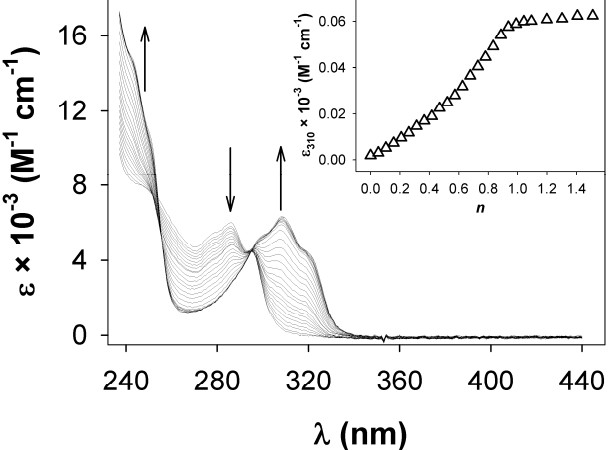

**Figure 2.** Absorption spectra recorded during the titration of a MeCN solution of **2a** ($10^{-5}$ M) with zinc(II) ion. In the inset, the titration profile shows the formation of a 1:1 adduct (*n* = equiv. of Zn$^{II}$ ion/equiv. of **2a**).

Considerable variations in the initial spectrum displayed by **2a** are observed: the band centered at 286 nm undergoes a distinct intensity decrease, while a new, poorly structured band arises at 308 nm. Two isosbestic points are initially observed at 255 and 295 nm but they are not preserved during the experiment. These changes are observed until 1 equiv. of metal ion is added; then, the spectral variation are almost undetectable. The spectral changes observed during the titration with zinc(II) can be ascribed to the interaction of the di(imino)pyridine system with the metal ion; a similar bathochromic shift has been observed in other iminopyridine ligands upon coordination with different metal ions [60,61].

The titration profile (see inset in Figure 2) obtained by plotting the molar absorbance (at 310 nm) versus $n$ (metal equiv./ligand equiv.) indicates the formation of a complex displaying a 1:1 stoichiometry as the main coordination species in solution. This species should involve, beside the tridentate di(imino)pyridine ligand, one or more solvent molecules in order to complete the coordination sphere of the metal ion, as zinc(II) could form tetra-, penta- or hexacoordinated species [62–66]. The formation of an octahedral 1:2 (metal/ligand) adduct cannot be ruled out, particularly at the beginning of the titration experiment when the metal/ligand equivalent ratio ($n$) is low; that could explain the inflection point observed in the titration profile at about $n = 0.5$.

The behavior of derivatives **4–8**, containing aromatic donor groups, during the titration experiments with zinc(II) is quite different from that displayed by **2a**. As an example, the spectra recorded during the titration of ligand **5b** ($10^{-5}$ M) with zinc trifluoromethanesulfonate in MeCN are reported in Figure 3.

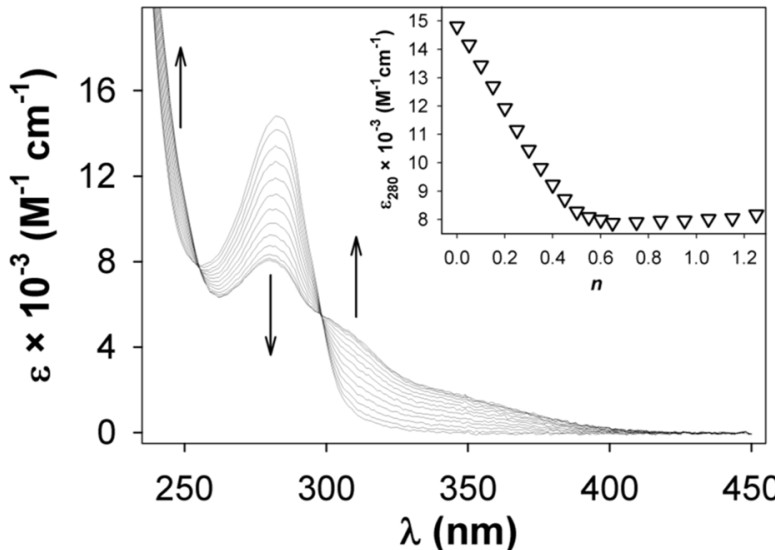

**Figure 3.** Absorption spectra recorded during the titration of an MeCN solution of **5b** ($10^{-5}$ M) with zinc(II) ion. In the inset, the titration profile shows the formation of a 1:2 adduct ($n =$ equiv. of $Zn^{II}$ ion/equiv. of **5b**).

The intensity of the band centered around 280 nm, upon the coordination to the metal ion, decreases as expected but this variation is observed until 0.5 equiv. of the metal ion are added, suggesting the formation of 1:2 adducts as the main, if not unique, coordination species in solution. The isosbestic points at 250 and 302 nm are well-defined and preserved during all the titration experiment, indicating that no other complex species form even in the presence of excess zinc(II).

In addition to the new band that develops at about 310 nm, a broad absorption tail is observed at the longer wavelength region (up to 400 nm). The titration profile (inset in Figure 3) also suggests the formation of the complex with 1:2 stoichiometry.

Similar results were obtained by titrating the other ligands **4–8** with zinc(II): the profiles resulting from the titration experiments clearly indicate that a 1:2 (metal/ligand)

complex species forms in each case. The shoulder or long-wavelength absorption tail appearing above 340 nm displays different intensity (and range) depending on the aromatic substituents and it is particularly intense in the case of the ligand **7a** ($\varepsilon$ = 3700 M$^{-1}$ cm$^{-1}$ at 350 nm) containing the 2,4,6-trimethoxyphenyl group (Figure 4). Families of spectra and the related titration profiles obtained for all the other di(imino)pyridine derivatives are reported in Supplementary Materials (SM).

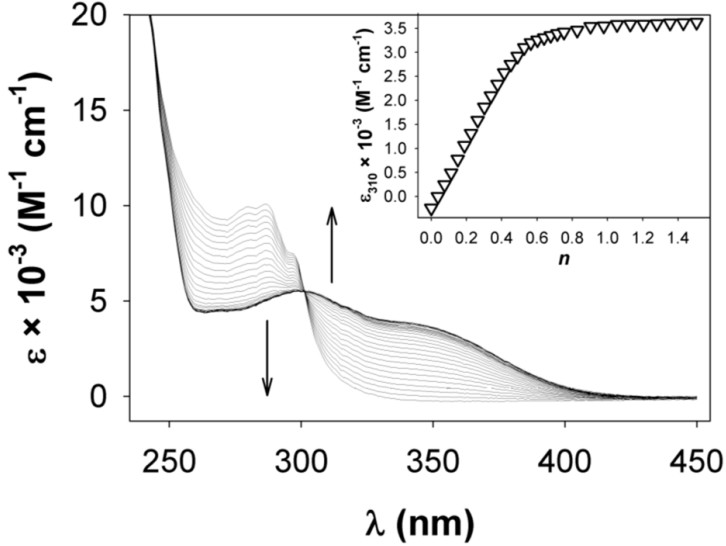

**Figure 4.** Absorption spectra recorded during the titration of a MeCN solution of **7a** ($10^{-5}$ M) with zinc(II) ion. In the inset, the titration profile shows the formation of a 1:2 adduct ($n$ = equiv. of Zn$^{II}$ ion/equiv. of **7a**).

Electrospray ionization mass spectroscopy (ESI-MS) experiments were also performed in order to gain a better insight on the identity of complex species formed after the addition of zinc(II) to a solution of ligand **7a** (see Figure S9). The mass spectrum of the free ligand in MeCN displays a base peak at $m/z$ = 494.1, corresponding to the [**7a** + H]$^+$ species. After the addition of 0.5 equiv. of Zn(CF$_3$SO$_3$)$_2$, a base peak at $m/z$ = 525 can be observed in addition to the peak ascribed to the plain ligand. The new peak corresponds to the [Zn(**7a**)$_2$]$^{2+}$ species, confirming the formation of 1:2 adducts. Interestingly, a similar mass spectrum is observed when more zinc(II) is added to the solution of **7a**: in particular, the peak at m/z= 525 represents the base peak both after adding 1 equiv. of Zn(CF$_3$SO$_3$)$_2$ and even in the presence of excess Zn$^{2+}$ (i.e., 2 equiv.). Possible species related to the hypothetical 1:1 adduct (for instance: $m/z$ = 278.6, [Zn(**7a**)]$^{2+}$; $m/z$ = 299.1, [Zn(**7a**)MeCN]$^{2+}$; $m/z$ = 319.6, [Zn(**7a**)(MeCN)$_2$]$^{2+}$; $m/z$ = 340.1, [Zn(**7a**)(MeCN)$_3$]$^{2+}$; $m/z$ = 706.1, [Zn(**7a**)(CF$_3$SO$_3$)]$^+$; $m/z$ = 747.1, [Zn(**7a**)(CF$_3$SO$_3$)(MeCN)]$^+$; $m/z$ = 788.2, [Zn(**7a**)(CF$_3$SO$_3$)(MeCN)$_2$]$^+$) were not detected nor when an equimolar amount of Zn$^{2+}$ was added or when the metal ion concentration exceeded that of **7a**. The outcomes of the ESI-MS experiments confirm that the 1:2 adduct is the predominant, if not unique, complex species induced by the interaction of zinc(II) with the di(imino)pyridine ligand bearing electron-rich 2,4,6-trimethoxybenzene moieties.

Compounds **3b** and **9a** behaved quite differently from the di(imino)pyridines described above.

The spectral changes observed during the addition of Zn(CF$_3$SO$_3$)$_2$ to a solution of **3b** indicate the formation of the 1:2 adduct as the main species in solution, although the spectra taken at $n$ > 0.5 (see Figure S1 in SM ) show slight but detectable variations, and the isosbestic points (at 297 and 259 nm) observed during the titration experiment are lost with the increasing metal-ion amount. This latter observation seems to suggest that the 1:1 complex might also form at higher metal-ion concentration levels (i.e., at zinc(II)/**3b** molar ratio exceeding 0.5). Moreover, due to the weak donor character of the phenyl groups, the arising band at 310 nm does not show the extended broad absorption tail at a

higher wavelength, as in the case of ligand **4–8** characterized by the presence of marked electron-donor aromatic groups.

Nitrophenyl derivative **9a** in MeCN shows broad absorption bands centered at about 280 nm, due to the overlapping of the bands expected for the di-(imino)pyridine and nitrophenyl chromophores. The titration with zinc(II) trifluoromethanesulfonate makes the intensity of these bands decrease until 1 equiv. of metal ion is added, while there is a less intense band increase at 260 and 325 nm, respectively (see Figure 5).

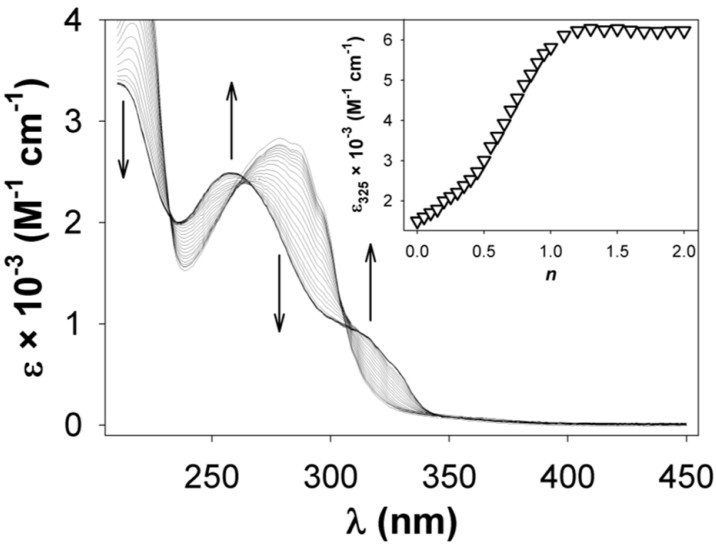

**Figure 5.** Absorption spectra recorded during the titration of an MeCN solution of **9a** ($10^{-5}$ M) with zinc(II) ion. In the inset, the titration profile shows the formation of a 1:1 adduct ($n$ = equiv. of Zn$^{II}$ ion/equiv. of **9a**).

The isosbestic points at 232, 262, and 303 nm (observed at the beginning of the titration) are progressively lost, indicating the presence of more than two species in solution. The titration profile shows also an inflection point at $n = 0.5$ (see inset in Figure 5), suggesting that a complex species corresponding to a 1:2 stoichiometric ratio can exist in solution in the presence of an excess of ligand. However, the UV-vis spectrum of $[Zn(3b)_2]^{2+}$ does not exhibit a clear and detectable low-energy shoulder as observed in the complexes formed by **4–8**.

The results obtained from the spectrophotometric titration experiments can be explained on the basis of the electronic properties of the different aromatic moieties connected to the di(imino)pyridine system. Ligands **4–8** contain aromatic substituents displaying an electron-donor character and form 1:2 (metal/ligand) octahedral complexes in the presence of a zinc(II) ion. The appearance of a broad shoulder during their spectrophotometric titrations with a metal ion is consistent with the formation of intramolecular EDA species and the occurrence of the corresponding CT transition as the metal–ligand interactions take place. Such an event, already reported for systems **1a** and **1b** [51], can be ascribed to an enhanced communication between the electron-rich aromatic moieties and the di(imino)pyridine system, which, following metal-ion coordination, becomes an electron-deficient unit analogous to the pyridinium ion [47,51]. Some examples of inter- and intramolecular EDA interactions involving pyridinium-ion derivatives and various aromatic hydrocarbon moieties have been reported [67–69].

On the other hand, these intracomplex EDA interactions are prevented in complexes (both 1:1 and 1:2) formed by ligand **2a**, as the aliphatic dodecyl chain does not have the right electronic features to be involved in EDA interaction with the coordinated di(imino)pyridine fragment. Therefore, no new band due to a CT transition is observed during the titration experiment involving this ligand.

The lack of the EDA interactions in the $[Zn(\mathbf{2a})_2]^{2+}$ complex can be also invoked to explain its lower stability if compared to the analogous complexes formed by ligands **4–8**. In fact, the 1:2 adduct $[Zn(\mathbf{L})_2]^{2+}$ (**L** = **4–8**), which is able to establish multiple EDA interactions beside the metal–ligand interactions, can benefit from an "extra-stability" [70] that makes it the only complex species even in the presence of metal-ion excess. On the contrary, $[Zn(\mathbf{2a})_2]^{2+}$ cannot profit from these intracomplex interactions and disassembles in the presence of excess zinc(II), affording the corresponding 1:1 complex.

The analysis of spectrophotometric data gives us a rough assessment of the stability constants corroborating the former suggestions. The titration with ligands **4–8** generally gives rise to only the 1:2 zinc complex species and the data can be fitted with only a $K_{1:2}$ of the order of magnitude of $10^{13}$; for instance, we found a $K_{1:2}$ value of about $4.0 \times 10^{13}$ and a value of $1.0 \times 10^{13}$ for the above-reported **5b** and **7a** ligands, respectively. Nicely, for **8b,** a $K_{1:2}$ with the same value of **5b** was obtained; in fact, both the ligands are featured by the same number of donor atoms in the same position onto the aromatic substituents.

On the contrary, for ligand **2a**, two stability constants, accordingly to the 1:1 and 1:2 complexes, were found to be $K_{1:1} = 3.2 \times 10^{10}$ ca. and $K_{1:2} = 3.2 \times 10^7$, which are, for example, more than $10^6$ times lower than that of the corresponding complex of **5b**.

An analogous explanation can be given for the behavior of ligand **9a** containing the electron-deficient nitrophenyl moiety. The constants of stability that were found are $K_{1:1} = 5.0 \times 10^8$ and $K_{1:2} = 2.5 \times 10^6$, which are even lower than that of the complex ligand $[Zn(\mathbf{2a})_2]^{2+}$.

The behavior of **3b** is intermediate and could be explained by taking into account the poor electron-donating character of phenyl substituents (if compared to the poly-methoxyphenyl groups present in ligands **4–8**), which could, in principle, establish only weak EDA interactions with the di(imino)pyridine fragment. Although it does not show the broad absorption tail of the EDA complexes of ligand **4–8,** and, from the above discussion, a 1:1 species could be expected, the calculated constant indicates only the formation of the 1:2 complex with a $K_{1:2} = 2.5 \times 10^{12}$. Therefore, the phenyl substituent is able to stabilize the 1:2 complex species, even if it is with an order of magnitude lower than that of compound **4–8**.

### 3.3. Crystallographic Studies

In order to have a better insight on the structural aspects concerning the EDA interactions taking place in the 1:2 adducts formed by the examined di(imino)pyridine ligands, we carried out crystallographic studies on some zinc(II) complexes. Suitable crystals of zinc(II) complexes with ligands **5b**, **7,a** and **8a** were obtained by the slow evaporation of MeOH solutions containing zinc(II) trifluoromethanesulfonate and the corresponding ligands.

Views of the molecular structures for the $[Zn^{II}(\mathbf{5b})_2]^{2+}$, $[Zn^{II}(\mathbf{7a})_2]^{2+}$, and $[Zn^{II}(\mathbf{8a})_2]^{2+}$ cation complexes, obtained as triflate salts, are shown in Figure 6, and selected geometrical features around the metal centers are reported in Table 3.

In all crystals, $Zn^{II}$ centers are fully coordinated by the six N atoms of two tridentate podand ligands placed in a meridional coordination geometry. The metal ion results in a distorted octahedral coordination because, as expected, the presence of a di(imino)pyridine moiety imposes a reduction in the axial $N_{imine}$-$Zn$-$N_{imine}$ angle from the ideal value of 180° [71]. The observed $N_{imine}$-$Zn$-$N_{imine}$ angles are in the range 148.5(2)–150.1(3)°.

For all molecular complexes, the mean $Zn$-$N_{pyridine}$ distances (2.064(8) Å for the **5b** ligand, 2.060(6) Å for the **7a** ligand, and 2.050(3) Å for the **8a** ligand) are similar, and all distances are shorter than the $Zn$-$N_{imine}$ distances, which are in the range 2.195(9)–2.304(3) Å. Moreover, the mean $Zn$-$N_{imine}$ distance in the $[Zn^{II}(\mathbf{5b})_2]^{2+}$ molecular cation, 2.210(8) Å, is significantly shorter than the $Zn$-$N_{imine}$ values observed in the other two molecular complexes: 2.263(4) Å for $[Zn(\mathbf{7a})_2]^{2+}$ and 2.259(4) for $[Zn(\mathbf{8a})_2]^{2+}$.

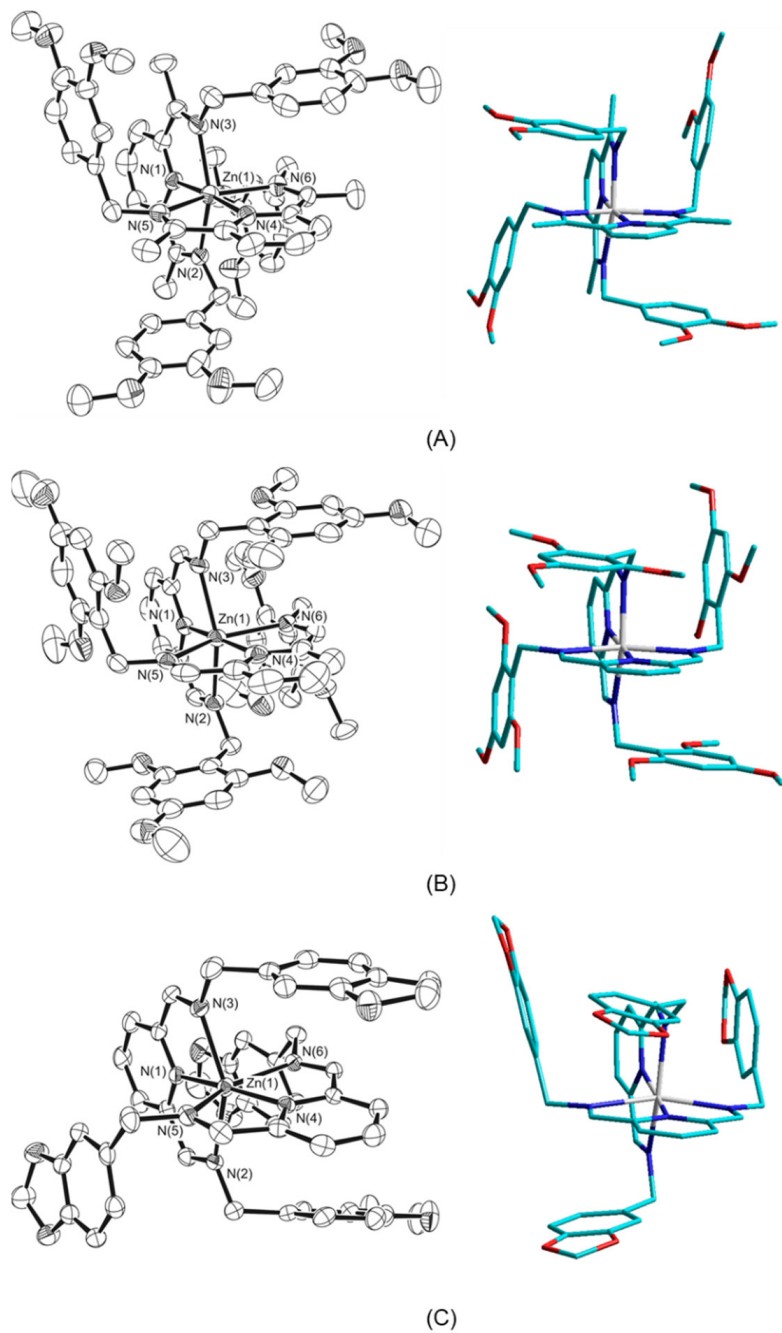

**Figure 6.** Plots showing thermal ellipsoids (left) and simplified sketches (right) of the three Zn$^{II}$ complexes: (**A**) [Zn(**5b**)$_2$](CF$_3$SO$_3$)$_2$·MeOH; (**B**) [Zn(**7a**)$_2$](CF$_3$SO$_3$)$_2$; (**C**) [Zn(**8a**)$_2$](CF$_3$SO$_3$)$_2$; ellipsoids are drawn at the 30% probability level; hydrogen atoms, triflate counterions and additional solvent molecule are omitted for clarity; atom symbols are shown only for nitrogen and metal center.

This shortening can be related to the presence of the -CH$_3$ group in the X position of the ligand (Figure 1). Actually, inspection of the Cambridge Structural Database (2022 release) [72] shows that, for octahedrally coordinated Zn(II) metal centers chelated by at least three N atoms of ligands based on a 2,6-di((1-imino)ethyl)pyridine moiety (as our **5b** ligand), the mean Zn-N$_{imine}$ distance is 2.22 Å and the mean Zn-N$_{pyridine}$ distance is 2.07 Å. When the three N atoms belong to ligands based on a 2,6-di((1-imino)methyl)pyridine moiety (as our **7a** and **8a** ligands), the mean Zn-N$_{imine}$ distance increases to 2.27 Å, whereas the mean Zn-N$_{pyridine}$ distance remains similar, being 2.04 Å.

**Table 3.** Selected geometrical features (bond lengths in Å and bond angles in °) around the metal center in the three studied crystals.

| | [Zn(5b)₂](CF₃SO₃)₂·MeOH | [Zn(7a)₂](CF₃SO₃)₂ | [Zn(8a)₂](CF₃SO₃)₂ |
|---|---|---|---|
| Zn(1)-N(1) | 2.059(8) | 2.049(5) | 2.053(3) |
| Zn(1)-N(4) | 2.068(8) | 2.071(6) | 2.047(3) |
| Zn(1)-N(2) | 2.237(8) | 2.282(4) | 2.304(3) |
| Zn(1)-N(3) | 2.195(9) | 2.261(4) | 2.225(4) |
| Zn(1)-N(5) | 2.209(8) | 2.245(5) | 2.230(4) |
| Zn(1)-N(6) | 2.201(8) | 2.264(4) | 2.275(3) |
| N(1)-Zn(1)-N(2) | 74.0(3) | 74.4(2) | 74.1(1) |
| N(1)-Zn(1)-N(3) | 75.6(3) | 75.3(2) | 75.5(1) |
| N(1)-Zn(1)-N(4) | 175.0(3) | 174.6(2) | 169.7(1) |
| N(1)-Zn(1)-N(5) | 103.4(3) | 109.5(2) | 113.1(1) |
| N(1)-Zn(1)-N(6) | 106.5(3) | 102.0(2) | 97.5(1) |
| N(2)-Zn(1)-N(3) | 149.6(3) | 149.7(2) | 148.7(1) |
| N(2)-Zn(1)-N(4) | 101.2(3) | 101.7(2) | 100.0(1) |
| N(2)-Zn(1)-N(5) | 91.2(3) | 92.8(2) | 92.1(1) |
| N(2)-Zn(1)-N(6) | 96.6(3) | 94.9(2) | 98.7(1) |
| N(3)-Zn(1)-N(4) | 109.1(3) | 108.6(2) | 111.2(1) |
| N(3)-Zn(1)-N(5) | 94.9(3) | 96.0(2) | 93.2(1) |
| N(3)-Zn(1)-N(6) | 92.8(3) | 92.5(2) | 92.2(1) |
| N(4)-Zn(1)-N(5) | 74.8(3) | 74.2(2) | 75.0(1) |
| N(4)-Zn(1)-N(6) | 75.3(3) | 74.3(2) | 74.9(1) |
| N(5)-Zn(1)-N(6) | 150.1(3) | 148.5(2) | 149.4(1) |

The two *mer*-terdentate ligands that bond each metal center are placed almost perpendicularly in all the three molecular cations; the dihedral angle between the di(imino)pyridine moieties are 86.4(2), 89.4(1) and 88.8(1)° for the **5b**, **7a** and **8a** ligands, respectively.

The pendant aromatic arms for the **5b** and **8a** ligands are placed according to a *trans* conformation, whereas the **8a** ligand results in a *cis* conformation. In spite of this difference, in all molecular complexes, each 2,6-di(imino)pyridine group is sandwiched between the two aromatic substituents of the other 2,6-di(imino)pyridine moiety chelates to the same metal center, and all complexes exhibit an orthogonally oriented molecular turn around the anchoring metal ion similar to those previously observed for other $Zn^{II}$ molecular complexes [73–75]. In particular, the dihedral angles between the pyridine rings and the adjacent two aromatic substituents (belonging to the other ligand moiety) are in the range 8.5(5)–10.5(4)° for $[Zn^{II}(\textbf{5b})_2]^{2+}$ and 4.4(2)–5.3(2)° for $[Zn(\textbf{7a})_2]^{2+}$. The range becomes 5.8(1)–28.4(1)° in the $[Zn(\textbf{8a})_2]^{2+}$ molecular cation, i.e., ligands in a *cis* conformation exhibit a minor parallelism between the aromatic rings.

The centroid–centroid separations between the di(imino)pyridine groups and the adjacent phenyl rings are in the range 4.07(1)–4.43(1) Å for $[Zn^{II}(\textbf{5b})_2]^{2+}$, 3.94(1)–4.37(1) Å for $[Zn(\textbf{7a})_2]^{2+}$ and 3.49(1)–4.18(1) Å for $[Zn(\textbf{8a})_2]^{2+}$ cationic complexes. These separations are significantly larger than observed in the complex formed by **1a** (3.35, 3.40 and 3.67 Å), [51] in particular for the $[Zn^{II}(\textbf{5b})_2]^{2+}$ and $[Zn(\textbf{7a})_2]^{2+}$ molecular cations. This could be due to the methoxy substituents on the aromatic moieties of **5b** and **7a**, whose steric (geometrical) hindrance could not allow a closer approach to the pyridine ring. These relatively large distances between the electron-rich and electron-deficient subunits seem to suggest that face-to-face EDA interactions between the aromatic rings could be not directly involved in the formation of CT species.

In fact, a careful examination of the molecular structures indicates that contacts between the aromatic rings of the pendant arms and the facing C=N groups of the di(imino)pyridine moieties could be favorable to EDA interactions. In particular, for zinc(II) complexes of **5b** and **7a** ligands, each of the four imino double bonds is overlapped by one aromatic ring and the distances between the centroid of the aromatic ring and the middle of the C=N group are 3.54(1), 3.58(1), 3.59(1), and 3.68(1) Å in $[Zn^{II}(\textbf{5b})_2]^{2+}$ and 3.50(1), 3.55 (1), 3.56(1), and 3.72(1) Å in $[Zn^{II}(\textbf{7a})_2]^{2+}$ molecular cations. Additionally, in

the [Zn$^{II}$(**8a**)$_2$]$^{2+}$ cationic complex, where the ligand results in a *cis* conformation, two of the four imino groups are faced with aromatic terminal groups, and the observed separations between the phenyl ring and the adjacent imino group are 3.50(1) and 3.69(1) Å.

### 3.4. NMR Studies

NMR experiments were also performed in order to better characterize the behavior of the di(imino)pyridine ligands in the solution containing a zinc(II) cation. In particular, $^1$H-NMR spectra of compound **6a** in CD$_3$CN were measured in the presence of increasing amounts of zinc(II) trifluoromethanesulfonate (see Figure 7).

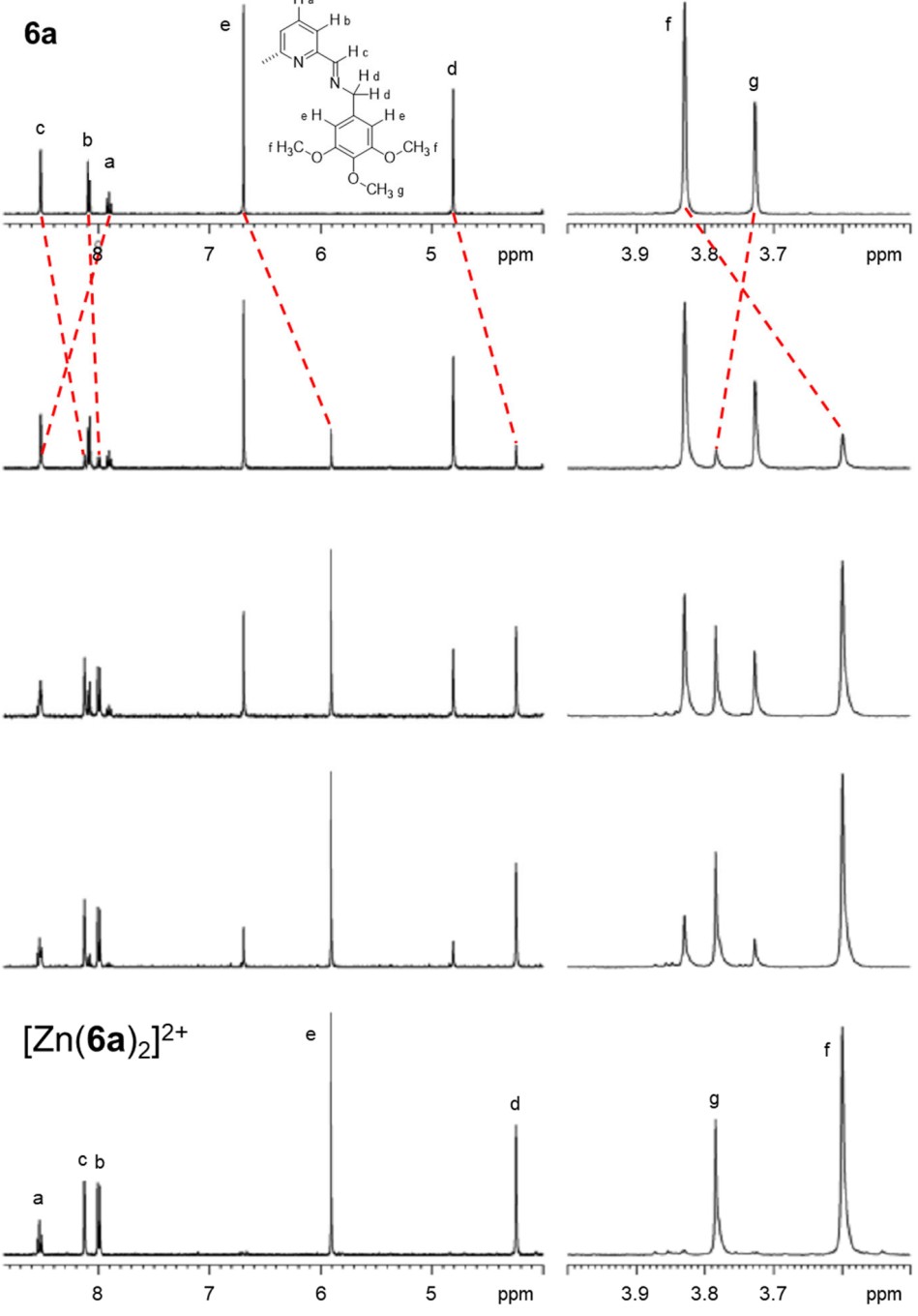

**Figure 7.** $^1$H-NMR spectra of a solution of **6a** in CD$_3$CN (25 °C, 5 × 10$^{-4}$ M) in the presence of increasing amounts of zinc(II) (0.00, 0.10, 0.25, 0.40, 0.50 equiv. from top to bottom).

The coordination of the metal ion according to a 1:2 (zinc(II)/ligand) stoichiometric ratio induces an upfield shift for all protons except for protons labelled as *g* and proton *a*, which undergo a downfield shift (from 3.73 to 3.78 ppm and from 7.90 to 8.53 ppm, respectively). All the chemical shifts are reported in Table S1 in the SM. These spectral variations can be explained by considering the combination of the effect due to the $Zn^{2+}$ ion, which alters the electron density of the nuclei and the anisotropic effects due to aromatic electron circulation. In particular, the upfield shift for protons *c*, *d* and *e* may be also provided by the reciprocal "face-to-face" arrangement of the pendant aromatic substituents with the imine moieties of the other ligand, corroborating the picture of the crystal structures in solution as well. No further detectable variations in the NMR peaks are observed by increasing the amount of zinc(II) in solution. This does further confirm the stability of the 1:2 complex in solution, when the terminal aromatic rings of the ligand have an electron-rich nature.

## 4. Conclusions

In this work, a novel series of 2,6-di(imino)pyridine ligands with different electron-rich aromatic substituents was prepared and the formation of complexes with the zinc(II) ion was investigated, in order to assess the generation of CT species promoted by metal-ion coordination.

Spectrophotometric titrations indicate the presence of the complex with a 1:2 (metal/ligand) stoichiometry as the main species in solution, even after the addition of an excess of zinc(II) ions, for ligands **4**–**8** featured by marked electron-donor methoxy groups on the aromatic substituents. This behavior is also confirmed by mass spectroscopy experiments. Moreover, a new band at about 310 nm with a board absorption tail (up to 400 nm) is observed. This new band can be attributed to the occurrence of a CT process involving interligand EDA interactions between the electron-donor aromatic substituents and the electron-acceptor di(imino)pyridine system coordinated to the metal center. Indeed, in the case of ligand **2a**, having two aliphatic chains, no EDA interactions can occur and, as a consequence, no CT band appears. Analogously, the ligand **9a** with the electron-acceptor nitrophenyl substituents is not able to generate the CT process. In both the latter cases, the 1:1 species is the most favorite complex with the zinc(II) ion in solution.

An intermediate situation is observed in the case of ligand **3b** with the simple phenyl substituents, having only a poor electron-donor character; a stable 1:2 complex was observed but no CT band was detected. Therefore, the latter substituent is able to stabilize the 1:2 complex packaging but is not sufficiently electron-donor in nature to give an EDA interaction.

The nature of the EDA interaction was confirmed by crystallographic studies, which disclose the electron-poor and electron-rich moieties involved in the CT process; the electron-donor aromatic rings are in a "face-to-face" arrangement with the $\pi$-system of the imino bonds. These moieties mutually belong to both the ligands and are forced in a favorable spatial arrangement by the coordinative preferences of the metal ion.

**Supplementary Materials:** The following supporting information can be downloaded at: https://www.mdpi.com/article/10.3390/chemistry4030051/s1, Figures S1–S8: absorption spectra and titration profile of compounds **3b**, **4a-b**, **5a**, **6a-b** and **8a-b**; Figure S9: ESI-MS spectra of the titration of compound **7a** with $Zn^{II}$ ion; Table S1: proton chemical shift of **6a** and $[Zn(6a)_2]^{2+}$.

**Author Contributions:** Conceptualization, M.L. and A.O.B..; investigation, C.C., G.C., M.B., M.L.W. and D.S.; resources, C.C. and G.C.; writing—original draft preparation, C.C., G.C. and M.B.; writing—review and editing, M.L. and A.O.B.; supervision, M.L. All authors have read and agreed to the published version of the manuscript.

**Funding:** This research received no external funding.

**Data Availability Statement:** The data presented in this research study are available in the present article and in the related Supplementary Materials.

**Acknowledgments:** The authors would like to gratefully acknowledge Enrico Monzani (University of Pavia) for his assistance in the NMR experiments.

**Conflicts of Interest:** The authors declare no conflict of interest.

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
