# Peer review of "Interligand Charge-Transfer Processes in Zinc Complexes"

_chemistry, doi:10.3390/chemistry4030051_

Round 1

Reviewer 1 Report

The paper by Licchelli et al. reports an interesting study concerning complexes of  Zn(II) with a series (12) of  2,6-di(imino)pyridine novel ligands, with different electron-rich aromatic substituents. The complexes have been characterized by absorption spectroscopy, X-ray diffraction, and ESI-MS. The stoichiometry of the complexes has been determined by absorption spectroscopy and titration data were used for determining the formation constants. Elemental analysis of the new compounds has also been reported.

The manuscript represents a well done application  of well known techniques and, although not containing relevant methodological improvements, in my opinion deserves to be published. Indeed, it is certainly well done, well written, potentially of great interest for the Chemistry readership, and reporting a sufficiently wide repertoire of references. Moreover the conclusions of the work are well supported by the presented data. I therefore recommend publication in the present form.

Author Response

We thank the Reviewer 1 for his kind comments and revision.

We carefully revised the whole manuscript in order to fix all the misspellings.

Reviewer 2 Report

The manuscript by Licchelli et al. reports the Inter-ligand charge-transfer processes in zinc complexes.  

This manuscript described a series of novel 2,6-di(imino)pyridine ligands with different electron-rich aromatic substituents and their 1:2 (metal/ligand) complexes with zinc(II) in which the formation of a CT species is promoted by the metal ion coordination. The absorption properties of these complexes have been studied, showing the presence of a CT absorption band, the nature of EDA interaction has been

confirmed by crystallographic studies, which disclose the electron-poor and electron-rich moieties

involved in the CT process. The experimental and characterization section are well conducted. 

I recommend publication with the corrections listed below.

I recommend publication of this manuscript in Chemistry subject to following modifications.

Page 3, section 2.2, please write coupling constant in 1HNMR of all compounds

Page 3, section 2.2, line 115, change 2,6 diacetylpiridine with 2,6 diacetylpyridine

Page4, 1HNMR 6b, 6H missing

Page5, 1HNMR of 9a, 1H extra

Page 8, line 294, change Fig1 with Fig 2.

Conclusion is missing in the manuscript, please write conclusion.

Author Response

We thank the Reviewer 2 for his carefully revision of our manuscript and we agree with his comments.

In the experimental part we added the couping constants in 1HNMR for all the compounds.

We revised the whole manuscript in order to fix all the misspellings.

The 1HNMR data of compunds 6b and 9a were corrected.

We changed Fig.1 with Fig.2 in the text as indicated by the Reviewer.

We added the conclusions paragraph.